# Network-based Active Inference for Adaptive and Cost-efficient Real-World Applications: A Benchmark Study of a Valve-turning Task Against Deep Reinforcement Learning

## Abstract

This paper introduces Network-based Active Inference (NetAIF), a novel approach that integrates Active Inference (AIF) principles with network dynamics to enable adaptive, cost-efficient real-world applications. In benchmark tests against Deep Reinforcement Learning (DRL), NetAIF outperforms DRL in both computational efficiency and task performance. Leveraging random attractor dynamics, NetAIF generates real-time trajectories, allowing robots to adapt to complex, dynamic environments without the need for extensive pre-training. We demonstrate NetAIF's superiority in industrial valve manipulation, achieving over 99% accuracy in goal position and orientation in untrained dynamic environments, with a 45,000-fold reduction in computational costs. NetAIF is approximately 100,000 times more efficient in iteration count than DRL, making it a highly robust and efficient solution for industrial applications.

## 1 Introduction

### 1.1 Overcoming Automation Challenges with Advanced Learning Methods

The World Energy Employment 2023 report by the IEA highlights a significant shift towards clean energy jobs, which now surpass fossil fuel employment, driven by a 40% rise in clean energy investment over the past two years. Despite economic and geopolitical challenges, the energy sector has seen growth in employment, particularly in solar PV, wind, EVs, and battery manufacturing. However, a shortage of skilled labor remains a key challenge, underscoring the need for targeted training and policy support to develop a workforce suited for the energy transition (IEA, 2023).

In response to these labor challenges, automation is playing an increasingly critical role in advancing the clean energy sector. Robotics, in particular, offers a promising solution to enhance operational efficiency and safety. However, to maximize the potential of robotics in complex and dynamic environments, sophisticated learning methods are required. One such approach, Deep Reinforcement Learning (DRL), has emerged as a leading candidate for enabling autonomous robotic systems in tasks like control, manipulation, and decision-making. Yet, despite its potential, DRL faces notable barriers to widespread adoption in the energy sector.

### 1.2 Deep Reinforcement Learning (DRL)

DRL combines the decision-making power of reinforcement learning (RL) with the pattern recognition capabilities of deep learning (DL). This allows robots to learn and adapt through trial and error, improving performance over time. DRL is increasingly explored for enabling autonomy in control and manipulation tasks in real-world environments by training agents to recognize complex patterns in data and make informed decisions.

However, DRL requires large amounts of data and time for agent training, as well as expert-designed reward functions to guide learning. Creating these reward functions demands substantial knowledge and engineering resources, as they must accurately capture desired outcomes, agent actions, and

constraints. Poorly defined reward functions can lead to suboptimal or unsafe behavior (Sutton & Barto, 2020). Thus, while powerful, DRL may not always be the most practical or cost-effective approach for every application.

### 1.3 AIF AS A NEXT GENERATION LEARNING METHOD

Active Inference (AIF) is a groundbreaking framework in neuroscience, offering a unified approach to understanding adaptive systems, including brain functions, and is gaining traction in fields like machine learning and robotics (Friston et al., 2006; Parr, 2019; Millidge, 2020; Lanillos et al., 2021). In robotics, AIF is reshaping control and learning by minimizing surprise rather than relying on reward-based mechanisms like DRL. Unlike DRL, which requires fixed environments, AIF utilizes a dynamic generative model, continuously adapting to changing surroundings through a feedback loop of prediction, perception, and action. This approach addresses the exploration-exploitation dilemma more fluidly by incorporating uncertainty directly into decision-making.

While AIF holds significant promise for creating adaptive robotic systems, its real-world deployment faces challenges due to the complexity of model design and high computational demands (Lanillos et al., 2021). Nonetheless, its potential to enhance flexibility, durability, and adaptability makes it a powerful alternative to traditional DRL techniques

### 1.4 NETWORK-BASED ACTIVE INFERENCE (NETAIF)

To overcome the limitations of both DRL and traditional AIF approaches, we propose Network-based Active Inference (NetAIF), a novel framework that leverages network dynamics to simplify trajectory calculations and enhance efficiency. Rooted in key AIF principles such as entropy and surprise minimization, NetAIF builds on the Free Energy Principle (FEP), which posits that systems self-organize by minimizing surprisal or prediction error. By harnessing the inherent dynamics of a network, NetAIF computes trajectories more efficiently than traditional AIF methods, reducing the need for complex mathematical models while enabling agents to adapt to dynamic environments in real-time. This streamlined approach makes NetAIF highly suitable for real-world robotic applications, offering significant improvements in both speed and computational cost.

## 2 NETWORK-BASED ACTIVE INFERENCE

### 2.1 NOTABLE CHARACTERISTICS

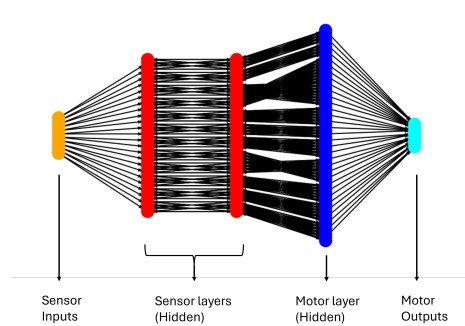

Figure 1: NetAIF network diagram for valve-turning task: parameters that determine the network structure such as number of layers, strides were determined through hyper parameter search

NetAIF's key innovation lies in its explicit feedback loops between hidden layers, which deliberately induce controlled instabilities to explore the state space more thoroughly (Brown, 2021)(Refer to Figs. 1 and 2). Unlike Recurrent Neural Networks (RNNs), where feedback is implicit (Mienye et al., 2024), NetAIF actively manipulates network dynamics to push the system into unstable regions. These feedback loops enhance oscillatory patterns, similar to neuron firing sequences, that

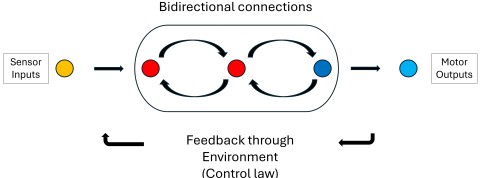

Figure 2: Bidirectional connection in hidden layers: the schematic diagram shows how the instability is induced within the hidden layer and how such instability is controlled via the external control law through feedback

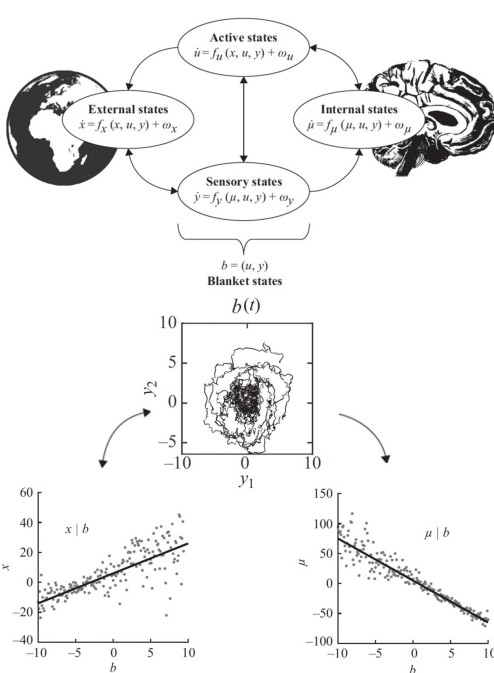

Figure 3: AIF brain and world - External states (world) are mirrored by internal states (brain). The active and sensory states (blanket states) couple external to internal states-rendering the system open. The (far from equilibrium steady-state) dynamics of each state is described with stochastic differential equations (*w* is a stochastic fluctuation). The images were adapted and modified from Parr et al. (2022)

persist even after training. This random bursts of node activity can be observed in the supplementary video, further highlighting the parallels with brain function. The introduction of these instabilities enables the system to maintain dynamic behaviors, known as itinerant (wandering) dynamics (Kaneko & Tsuda, 2003; Friston & Ao, 2012), allowing it to continuously adapt to changing environments.

NetAIF operates within the framework of Active Inference, where a system interacts with its environment through blanket states. Blanket states consist of sensory states, which gather external information, and active states, which influence the environment as shown in Fig.3. This dynamic interaction forms the core of the system's ability to operate in a Non-Equilibrium Steady State (NESS). In NESS, the system is never fully at rest but continuously adapts to changing inputs from the environment, minimizing prediction errors in real time. The feedback between sensory and active states ensures that the system remains stable yet flexible, adjusting its actions and beliefs to maintain optimal performance even in uncertain or complex environments. This aligns with Bayesian inference principles, as NetAIF constantly updates its beliefs in response to new sensory inputs and envi-

ronmental changes, enhancing its ability to navigate complex environments and discover optimal trajectories.

NetAIF also replaces traditional activation functions with a discrete weight-assigning mechanism, designed to reset node weights and maintain NESS. By leveraging the constant interaction between sensory and active states, NetAIF remains in a state of continuous exploration, avoiding local minima and ensuring that it adapts dynamically to new challenges. This stochastic function enhances the network's ability to explore different states, preventing it from being trapped in local optima.

Additionally, NetAIF integrates learning and control, guiding motor outputs with clear task-specific control laws. These laws break tasks down into sub-goals, such as aligning objects, allowing even non-experts to define behaviors without deep control theory knowledge. For instance, in a valve manipulation task, control instructions guide the network to minimize errors by aligning the vector of the valve's position with the one of the end effector. This ensures precise orientation and movement, making the system more intuitive and effective for real-world applications. This user-friendly approach facilitates seamless integration of learning and control.

---

**Algorithm 1** Main loop of the NetAIF model

1: **Initialize** all model parameters and weights
2: **while** system is running **do**
3:     Prediction_Error = $Desired\_State - Current\_State$
4:     Input_signals = $Prediction\_Error$
5:     **for** each weight $w$ in all weights **do**
6:         **if** magnitude of associated signal > threshold **then**
7:             Set $w = new\_weight\_value()$
8:         **end if**
9:     **end for**
10:     Input_to_hidden = $Input\_signals \times W\_input\_hidden$
11:     Feedback = $Hidden\_signals\_prev \times W\_hidden\_hidden$
12:     Hidden_signals = $Input\_to\_hidden + Feedback$
13:     Hidden_signals_prev = $Hidden\_signals$
14:     Outputs = $Hidden\_signals \times W\_hidden\_output$
15:     Motor_Commands = $Outputs$
16:     Send motor commands to actuators
17: **end while**

---

The core of the NetAIF framework is outlined in Algorithm 1. Each cycle calculates the prediction error between current and desired states, which updates network weights dynamically. If a signal exceeds a set threshold, its weight is reset to ensure stability. Feedback loops in the hidden layers facilitate adaptive behavior and robust trajectory generation. Motor commands are derived from the hidden layers and sent to the actuators, enabling real-time adjustments. This continuous feedback allows NetAIF to quickly adapt to changing environments, making it ideal for dynamic tasks like PV panel inspection.

## 2.2 THE RANDOM ATTRACTOR

To represent the NESS behavior in NetAIF, Random Dynamical Systems (RDS) are employed, providing a framework to understand complex systems driven by stochastic processes. In particular, random pullback attractors (Caraballo & Han, 2016), also known as stochastic basins of attraction, describe how NetAIF's state evolves over time in response to environmental uncertainty. Expressed as $\varphi(t, \omega, x)$, where $t$ is time, $\omega$ represents randomness, and $x$ is the state variable, these attractors characterize regions in the state space where the system tends to settle. The random attractor $\mathcal{A}(\omega)$ pulls trajectories towards it, ensuring that NetAIF remains adaptive and stable within its NESS framework, despite external randomness.

This is formalized by:

$$\lim_{t \to \infty} \text{dist}\left(\varphi(t, \theta_{-t}\omega, B), \mathcal{A}(\omega)\right) = 0$$

where $\theta_{-t}\omega$ is time-shifted random noise, and $B$ is a bounded set of initial conditions. This equation captures the system's tendency to converge towards the random attractor, influenced by past random events.

This convergence can be understood as stochastic diffusion in parameter space, driven by increasing random fluctuations in parameters (e.g., network connection weights) in high free energy regions. As the system nears free energy minima, random fluctuations decrease, stabilizing the trajectory. The

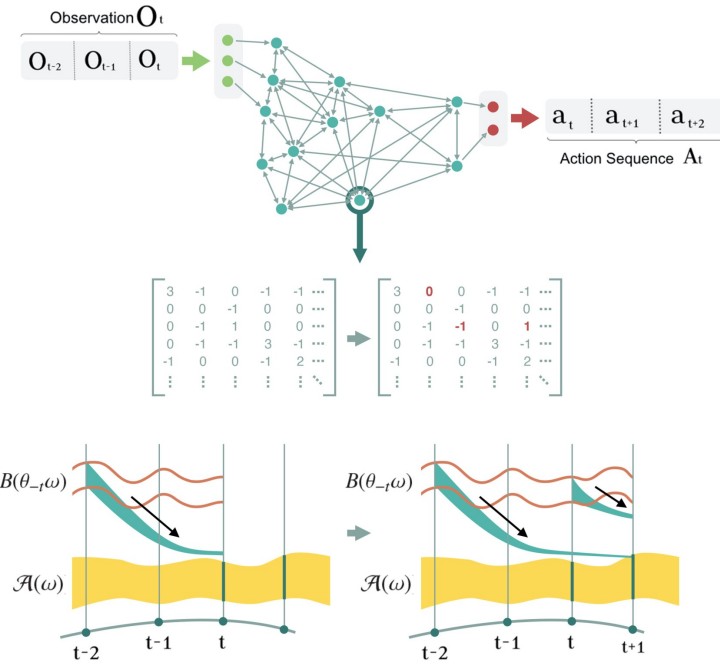

Figure 4: Abstract representation of a random pullback attractor, $\mathcal{A}$, and the random set, $B$. While the weights of the network are updated randomly (shown in matrix format), a flow from the random set emerges and gets attracted to the attractor.

dynamics can be described by a stochastic differential equation (SDE), like the Langevin equation:

$$dx = -\nabla F(x)\, dt + \sqrt{2\Gamma}\, dW$$

where $x$ represents system parameters, $F(x)$ is the free energy landscape, $\Gamma$ is the diffusion coefficient, and $W$ is a Wiener process. This equation reflects the balance between deterministic drift towards free energy minima and stochastic exploration of the parameter space, shaping the system's trajectory.

It is worth noting that NetAIF's optimization process is inherently local, as free energy is an extensive quantity—the system's total free energy is the sum of the free energies of its individual components. The variational free energy, which approximates the true free energy, is calculated using local prediction errors. Certain predictions are clamped with high precision or influenced by desired outcomes, defining the attracting set that represents the target state or desired sensor inputs. By minimizing local prediction errors, the network is guided toward this attracting set.

This local approach allows the system to efficiently navigate the free energy landscape without needing global computations or information propagation across the entire network. Iterative updates based on local prediction errors and control laws enable the system to converge toward desired states.

The roots of this learning process trace back to early cybernetics (Ashby, 1947; 1956) and stochastic thermodynamics (Ao, 2008; Seifert, 2012), where systems self-organize by reducing prediction errors. In NetAIF, this drives the network towards stable, efficient behavior, enabling smooth and precise robotic movements.

## 3  APPLICATIONS: VALVE MANIPULATION

### 3.1  PROJECT SCOPE

Manipulating valves is a common task in industrial environments, making it an ideal test case for evaluating NetAIF's capabilities in robotic manipulation and directly comparing it with DRL. The

results demonstrate NetAIF's accuracy in dynamic environments, reduced training data requirements, and lower computational costs. In this study, a robotic arm was tasked with:

- Turning valves of three shapes (square, circular, triangular) to a 45-degree clockwise angle.

- Manipulating valves without prior knowledge of their shape.

- Handling valves with varying resistances and slipperiness.

Although this study focuses on valve manipulation, NetAIF's flexible framework extends to other applications such as pick-and-place operations and assembly tasks, with future experiments planned to demonstrate its generalizability across various real-world scenarios.

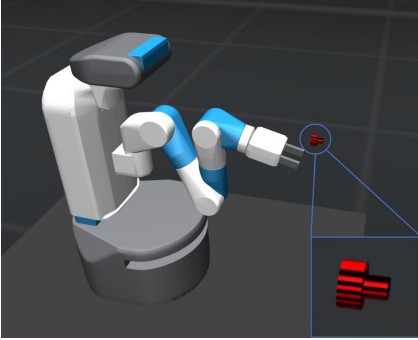

Figure 5: Valve-turning simulation environment (Mujoco)

## 3.2 MODEL ASSESSMENTS

### 3.2.1 EXPERIMENT 1: BASELINE EVALUATION AND ALGORITHM COMPARISON

To enable accurate comparisons of DRL algorithms, we developed a custom OpenAI gym environment for robotic valve manipulation, based on the FetchReach simulation. This environment includes a digital twin of the Fetch Mobile Manipulator, though control is focused solely on the manipulator arm (Fig. 5). This setup provided a consistent baseline for refining valve manipulation skills and comparing DRL algorithms. We fine-tuned hyperparameters and observation spaces for optimal performance and used Ornstein-Uhlenbeck noise (Hollenstein et al., 2023) to improve exploration, gradually adjusting it to enhance model performance, ensuring robust comparisons.

Nonetheless, given the difficulties DRL encounters with the intricate valve-turning operation, our analysis concentrated on a more basic variant of the task. Unlike the full task version, which begins with the arm in a neutral position and extends to engage the valve, this simplified version of the task initiates with the robotic arm already clasping the valve. Additionally, this version employs a shorter episode length, contributing to an enhanced rate of learning efficiency.

The performance outcomes of various algorithms on different valve types are summarized in Table 1. Our findings revealed that the square valve was the easiest to manipulate, followed by the triangular valve, while the circular valve was the most challenging due to its tendency to slip. Models using action noise often experienced training crashes with the circular valve. Performance was measured in terms of manipulation precision (error from target angle) and training cost, expressed in teraflops, focusing on the computational expense of model training.

In summary, the Soft Actor-Critic (SAC) (Haarnoja et al., 2019) algorithm showed promise but struggled with generalization, especially when valve shapes were randomized. Most SAC models failed during training, with only two surviving but unable to generalize well. Adding Hindsight Experience Replay (HER) (Andrychowicz et al., 2018) to SAC improved performance, but this combination still lacked the generalization needed. To address these challenges, we explored Proximal Policy Optimization (PPO) (Schulman et al., 2017) in the next experiment.

Table 1: Comparative results of DRL algorithms (Experiment 1)

| Component | SAC | SAC + HER | SAC + noise | SAC + HER + noise |
|---|---|---|---|---|
| Square(Reward) | -58.9 | -198.9 | -478.9 | -50.8 |
| Square(Errors) | 5.83° | 1.84° | 35.39° | 0.78° |
| Triangle(Reward) | -49.1 | -158.3 | -107 | -58.9 |
| Triangle(Errors) | 1.32° | 8.94° | 1.38° | 0.293° |
| Circle(Reward) | -81.3 | -424.1 | N/A | N/A |
| Circle(Errors) | 4.1° | 42.2° | N/A | N/A |
| Training Iterations | $1.5 \times 10^7$ | $1.5 \times 10^7$ | $1.5 \times 10^7$ | $1.5 \times 10^7$ |
| TFLOPs | 1434.9 | 1433.8 | 1441.9 | 1433.8 |
| Model Size | 3.4 MB | 3.4 MB | 3.4 MB | 3.4 MB |

### 3.2.2 EXPERIMENT 2: GENERALIZATION AND ROBUSTNESS WITH VARYING VALVE SHAPES AND PHYSICAL PROPERTIES

In the subsequent experiment, our research focused on evaluating the models' generalization capabilities by introducing a new valve shape in each episode, simulating real-world scenarios where valve shapes are unknown. We also tested the models' robustness by modifying physical properties such as friction and valve size, employing domain randomization to improve the adaptability of DRL models. However, despite limiting training to a practical duration of 3-4 weeks, many SAC and HER+SAC models struggled, largely due to the extensive datasets required by domain randomization. Only a few models successfully completed the training process.

Our evaluation revealed PPO as the most efficient DRL algorithm, demonstrating high rewards and accuracy across different valve shapes—square, triangular, and circular—highlighting its adaptability. However, performance variability across models indicated that multiple training iterations are often needed to find optimal strategies, reflecting the inherent unpredictability of DRL.

While this comparison was limited to a simplified task, as DRL struggled with the full task complexity, NetAIF demonstrated faster learning and greater robustness, outperforming all DRL algorithms in both simplified and full tasks. NetAIF's superior adaptability and consistency in dynamic environments, particularly under domain randomization, variations in physical properties, and reduced contact friction, emphasize its potential for real-world applications with low environmental predictability, showcasing its efficiency and adaptability in complex task execution.

Table 2: Comparative results of DRL algorithms (Experiment 2)

| Metric | PPO | SAC | HER+SAC | NetAIF |
|---|---|---|---|---|
| Reward | -35.2 | -307 | -181.4 | N/A |
| Errors | 2.7° | 19.7° | 14.1° | 0.131° |
| Training Iterations | $2.0 \times 10^8$ | $5.0 \times 10^7$ | $5.0 \times 10^7$ | 2,000 |
| TFLOPs | 749.8 | 4783.0 | 4779.3 | 0.016 |
| Model Size | 175KB | 3.4MB | 3.4MB | 16KB |

### 3.3 EFFICIENCY AND COST

The analysis reveals a stark contrast between NetAIF and leading DRL algorithms like PPO. NetAIF is about 100,000 times more efficient in iteration count, slashing computational costs by 45,000 times (in TFLOPs), and cutting training time by 99.99%, all while maintaining over 99% accuracy in adjusting the valve to a precise 45-degree angle.

For DRL to match NetAIF's performance, vastly more data and computing power would be required, significantly increasing costs and infrastructure demands, along with exponentially longer training times. In contrast, NetAIF offers a streamlined, cost-effective solution, enabling rapid training and deployment on devices with limited computing capacity.

## 3.4 Accuracy and Robustness

When the robustness of the DRL models was scrutinized post-training, it became apparent that their performance waned significantly when the test conditions deviated from those of the training scenario. Our robustness trials, which included altering physical attributes such as contact friction between valve and gripper, valve mass, and rotational damping, as well as output motor torque and decreasing contact friction, revealed a marked performance decline. Fig. 6 illustrates the decline in performance for each DRL algorithm as well as for NetAIF. For brevity, only the results pertaining to the circular valve shape are presented.

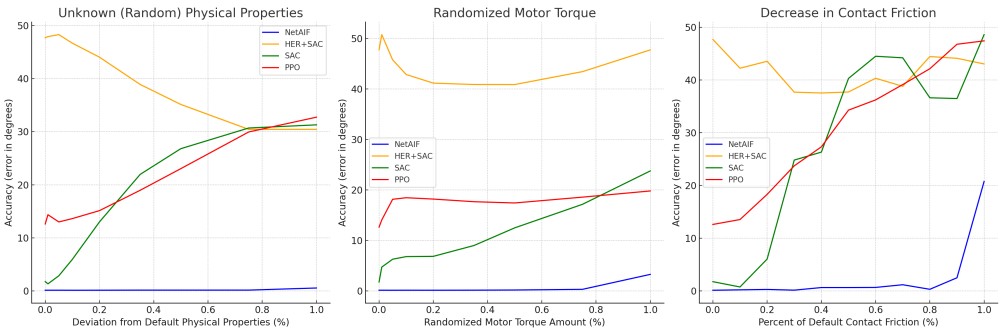

Figure 6: Valve-turning simulation (valve shape: circle) - left: unknown (random) physical properties; middle: randomized motor torque; right: decrease in contact friction

Table 3 details the performance reduction for the most proficient DRL model, PPO, highlighting the decline in accuracy levels for each category at deviations set to 75% of the default value. In stark contrast, the NetAIF model emerged as a paragon of generalization capability, seamlessly adapting to all valve shapes without reliance on domain randomization or additional tuning. Its robustness is validated by a steadfast accuracy rate exceeding 99% for square, triangular, and circular valves, as Table 4 illustrates. This is maintained across all test cases except in the extreme scenario of nearly non-existent contact friction, as Fig. 6 delineates. It is noteworthy that the simulation imposes a threshold at 1% contact friction between the valve and the gripper for operational stability.

Table 3: PPO turn accuracy (errors are measured as the deviation from the desired angle (i.e., $45°$) - Case A: unknown (random) physical properties; Case B: randomized motor torque; Case C: decreased contact friction; All the values are for 75% deviation from the default values

|          | Default        | Case A          | Case B          | Case C           |
|----------|----------------|-----------------|-----------------|------------------|
| Square   | $1.49°(96.7\%)$  | $7.10°(84.2\%)$   | $10.81°(76.0\%)$  | $35.17°(21.8\%)$   |
| Triangle | $7.20°(84.0\%)$  | $19.86°(55.9\%)$  | $22.56°(49.9\%)$  | $54.78°(121.7\%)$  |
| Circle   | $12.61°(72.0\%)$ | $29.95°(33.4\%)$  | $18.56°(58.8\%)$  | $46.80°(104.0\%)$  |

Table 4: NetAIF turn accuracy - The same criteria as the PPO above

|          | Default          | Case A           | Case B           | Case C            |
|----------|------------------|------------------|------------------|-------------------|
| Square   | $0.103°(99.8\%)$   | $0.097°(99.8\%)$   | $0.273°(99.4\%)$   | $0.097°(99.8\%)$    |
| Triangle | $0.119°(99.7\%)$   | $0.137°(99.7\%)$   | $0.364°(99.2\%)$   | $0.137°(99.7\%)$    |
| Circle   | $0.125°(99.7\%)$   | $0.147°(99.7\%)$   | $0.302°(99.3\%)$   | $0.1465°(99.7\%)$   |

### 3.4.1 Experiment 3: Robustness Testing in Real-World Scenarios

While the previous experiments demonstrated the efficiency and accuracy of NetAIF compared to DRL, we wanted to push the boundaries of NetAIF's robustness in more challenging, real-world conditions. In this third experiment, we focused on testing the model's ability to handle unpredictable environmental changes, such as faulty or inexpensive sensors and unexpected valve movements. Given the complexity of these scenarios, we opted not to assess DRL's performance, assuming the results would likely be unsatisfactory under these conditions.

The robustness testing focused on dynamically varying the valve's location and orientation through sinusoidal motion, random walks, and progressively increasing levels of sensor noise. This approach simulates real-world conditions where sensor-reported data, such as valve position and orientation, can deviate from actual values. To improve the model's resilience to noise and ensure stable performance under these uncertainties, we applied both a Kalman filter and a low-pass filter—commonly used techniques in practical robotic systems.

The results of these robustness tests, highlighting NetAIF's ability to maintain accuracy and stability in dynamic environments, are summarized in Tables 5 and 6.

Table 5: Model tolerance to environment changes while maintaining 95% accuracy

| Test | Max Tolerated |
|------|---------------|
| Valve Location (Circular motion) | 0.05 m at 0.1 Hz |
| Valve Orientation (Circular motion) | 20 degrees at 0.1 Hz |
| Valve Location (Random walk) | 0.2 m/sec |
| Valve Orientation (Random walk) | 20 degree/sec |

Table 6: Model tolerance to sensor noise while maintaining 95% accuracy - Case A: No filtering; Case B: Kalman filtering; Case C: Kalman and low-pass filtering

| Test | Case A | Case B | Case C |
|------|--------|--------|--------|
| Valve Location | 0.5 cm | 1 cm | 10 cm |
| Valve Orientation | 2 degrees | 5 degrees | 20 degrees |

## 4  REAL-TIME PERFORMANCE METRICS

Table 7 highlights the compact nature of the NetAIF model, evaluated on a real-world valve-turning task using the Lite6 robot from UFactory as shown on Fig. 8. The evaluation was conducted on an 8-core Intel Core i9 (I9-9880H) 2.4 GHz processor without GPU support. The network's update cycle is approximately 5ms, as detailed in Table 8, resulting in a remarkably short training time of just about 7 seconds for the valve-turning task. Once the network is trained, the resulting trajectory values become smoother with relatively small random fluctuations. Additionally, the model's stored weight values improve deployment flexibility, making it easily transferable and deployable across different systems. This portability ensures that similar tasks can be executed efficiently without requiring retraining, providing a key advantage—especially when the network is scaled up to handle more complex tasks.

The swift and efficient performance of the NetAIF model can be attributed to its FEP-guided path generation combined with random attractor dynamics. As illustrated in Fig. 7, these random attractor dynamics replace traditional motion planning components. Unlike some of conventional methods that require pre-calculated or trained trajectories, NetAIF generates the trajectory in real-time by continuously feeding sensor data to the random attractor, enabling flexible and adaptive motion planning.

Table 7: NetAIF model metric for valve-turning task

| Metric | Valve-turning |
|--------|---------------|
| Network Size (No. of Nodes) | 332 |
| Network Size (No. of Connections) | 1872 |
| Network Size (No. of Bytes) | 16304 |
| No. of Iterations to Convergence | 1413 |

The total motion planning time for the real world valve-turning task is summarized in Table 8. In comparsion, conventional planning algorithms such as PRM and Hybrid RRT-PRM can take up to 482 milliseconds in similar environments, largely due to the computational burden of recalculating

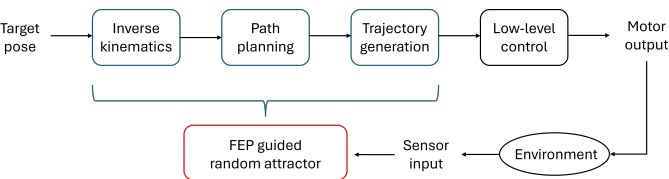

Figure 7: Motion planning process

paths (Jermyn, 2021). Research on dynamic systems, such as UAV-based models with visual processing, reports planning times between 50 and 500 milliseconds in dynamic environments (Cui et al., 2022). The standard deviation of 2.09 milliseconds indicates relatively low variability in planning times, underscoring the stability of the NetAIF model under varying valve dynamics.

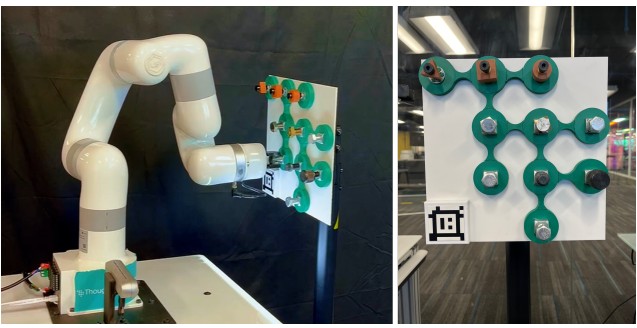

Figure 8: Valve-turning experiment setup. Left: The Lite6 robotic arm is used to manipulate valves of different shapes, while the Intel RealSense D455 camera provides valve localization. Right: Examples of valve shapes (triangle, square, and circle) and various bolts used in the experiments.

Table 8: Summary of time taken to generate values by the network

| Statistic | Value (milliseconds) |
|---|---|
| Mean time | 4.53 |
| Standard deviation | 2.09 |
| Median time (50th percentile) | 5.38 |
| 25th percentile | 2.75 |
| 75th percentile | 6.21 |

## 5 CONCLUSION

This paper presents Network-based Active Inference (NetAIF), a novel approach that combines AIF principles with network dynamics to surpass Deep Reinforcement Learning (DRL) in real-world robotic tasks. In an industrial valve manipulation benchmark, NetAIF achieved over 99% accuracy while reducing computational costs by 45,000 times and training time by 99.99%. Unlike DRL, which requires extensive pre-training, NetAIF leverages random attractor dynamics for real-time trajectory generation, adapting swiftly to dynamic environments with minimal overhead. Its robustness across varied conditions makes it a scalable, cost-efficient solution for adaptive robotic control in unpredictable settings.

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
