# OpenReview forum: "Network-based Active Inference for Adaptive and Cost-efficient Real-World Applications: A Benchmark Study of a Valve-turning Task Against Deep Reinforcement Learning"
_ICLR.cc/2025/Conference — ICLR 2025 Conference Desk Rejected Submission_

### Official Review · Reviewer_xMnq · 2024-10-25

**Soundness:** 3
**Presentation:** 2
**Contribution:** 3
**Rating:** 5
**Confidence:** 4

**Summary:**

A Neural Network incorporating Active Inference principles is applied in robotic tasks. With the aid of random attractor dynamics, an interesting portion of the presented approach is that no pre-training is forcibly required. The adaptability in dynamic environments with minimal overhead makes it a scalable, cost-efficient solution for robotic control in unpredictable settings.

**Strengths:**

This is an application of NetAIF in robotic control. Two main advantages are illustrated. One is that the Time-consuming pre-training is no longer required, making it a cost-efficient solution. Another is the incorporation of feedback control law in the network model, which enables the system stable. Moreover, random pullback attractor is utilized to update the weights. From the results in the paper, the performance of this method is very good.

**Weaknesses:**

The literature research is not enough and the literature is so old. The expression of the paper is weak and the key issues are not explained clearly. For example, lack of detailed application description of NetAIF, like noise processing. In addition, the arranged experiments are too simple. The motion speed of the manipulator is low, which is not enough to verify the effectiveness of the method.

**Questions:**

1)	Please provide the related code;
2)	I wonder if this method is effective for learning highly dynamic systems like quadcopters?
3)	Provide more experimental details, such as network input, network output, neuron weights, and the design of feedback controller.

---

> ### Author Response · Authors · 2024-11-18
>
> We thank the reviewer for their thoughtful comments and for highlighting the strengths of our work, particularly NetAIF’s elimination of time-consuming pre-training, cost-efficiency, and the incorporation of feedback control and random attractor dynamics. Below, we address the specific concerns raised and provide clarifications to strengthen the manuscript.
>
> First, regarding the perceived simplicity of the experiments, we emphasize that the valve-turning task was deliberately chosen as a representative industrial benchmark. This task is widely recognized in robotics as a meaningful challenge, requiring precise manipulation, adaptability to diverse valve shapes (e.g., circular, triangular, square), and robustness against varying frictional and resistance properties. These variations introduce significant complexity, ensuring that the experiments thoroughly evaluate NetAIF’s ability to adapt, learn, and perform robustly in real-world-like dynamic scenarios. The manipulator’s speed was set to the default configuration of the robotic platform, ensuring consistency and reproducibility across all comparisons, particularly with RL baselines. While the default speed may appear slow, it was sufficient to highlight NetAIF’s advantages in trajectory generation, accuracy, and computational efficiency. We will revise the manuscript to emphasize that NetAIF’s real-time trajectory generation capabilities inherently support high-speed and more dynamic tasks, which will be explored in future work.
>
> Regarding the literature review, we acknowledge the need to better situate NetAIF within the context of existing research. While the paper focuses on demonstrating the practical advantages of NetAIF over RL approaches, we agree that a discussion of prior work on neural network architectures, Active Inference-based methods, and reinforcement learning techniques would help clarify the novelty and contributions of our approach. In the revised manuscript, we will add a related work section, specifically addressing recent advances in these areas and how NetAIF compares to other models.
>
> The reviewer also requested additional experimental details, such as network input, network output, neuron weights, and the design of the feedback controller. We appreciate this suggestion and will include a more detailed explanation of the experimental setup in the revised manuscript. This will include the design of the feedback mechanism, the structure of the random attractor dynamics, and a breakdown of the network’s inputs and outputs. These details will make the technical contributions of NetAIF more accessible and strengthen the rigor of our work.
>
> Finally, we appreciate the reviewer’s question regarding the applicability of NetAIF to highly dynamic systems such as quadcopters. While the focus of this work was on industrial valve manipulation as a representative task, we believe NetAIF’s adaptability and real-time processing capabilities make it well-suited for high-dynamic scenarios. We will include a discussion in the manuscript about extending NetAIF to such systems, supported by preliminary observations where applicable.
>
> In summary, we will revise the manuscript to address these concerns, providing more context, technical detail, and clarity about the experimental design and its broader implications. Thank you for your constructive feedback, which has helped us identify ways to refine and improve the presentation of our work.

---

> > ### Comment · Reviewer_xMnq · 2024-11-28
> >
> > I thank the authors for their responses.   Question 1 and 2 have not yet been considered. In addition, I have opened the revised manuscript several times in the last few days, but I have not found the revised part you replied to question 3.  Please distinguish the colours in the revised manuscript.  I will update my score after all issues is addressed, especially the slow motion speed of the manipulator. It would better that some references are provided to support the importance of this task in industrial applications, even in default manipulator's speed.

---

### Official Review · Reviewer_TcU3 · 2024-10-26

**Soundness:** 2
**Presentation:** 1
**Contribution:** 3
**Rating:** 3
**Confidence:** 3

**Summary:**

The paper proposes Network-based Active Inference (NetAIF), a novel framework that integrates Active Inference (AIF) principles with network dynamics to create a real-time adaptive system for robotic tasks. Unlike reinforcement learning, which typically relies on learning from reward signals, NetAIF leverages feedback loops between hidden layers and actively learns by minimizing the difference between the current state and the desired state, enhancing the system's learning efficiency. The paper demonstrates the effectiveness of the proposed method through both simulation and real-world experiments, where NetAIF dramatically outperforms RL methods in terms of efficiency and accuracy.

**Strengths:**

1. The proposed method seems novel, NetAIF's ability to learn in real-time without requiring extensive pre-training offers a significant advantage over traditional RL methods.
2. The paper evaluates NetAIF on both simulated and real-world tasks, providing evidence of the method's practicality.

**Weaknesses:**

1. The paper is difficult to process, particularly for readers without a background in Active Inference. A major revision could improve the paper’s accessibility, especially for a general machine learning audience. For instance, a detailed explanation of how NetAIF differs from standard neural networks (particularly in weight updates compared to backpropagation) is necessary to help readers better understand the novel aspects of the framework.
2. There is no related work section, making it difficult to situate the contribution within existing literature. Adding a discussion on how NetAIF compares to other models, especially in terms of neural network architectures, reinforcement learning approaches, and AIF-based methods, would clarify the innovation.
3. It is unclear whether NetAIF is leveraging additional information compared to model-free RL algorithms. The paper should clarify whether NetAIF has access to more privileged information (e.g., task-specific control laws or goal states) and whether the comparison with RL is fair. Are RL algorithms using the same input data for training and execution?

**Questions:**

1. The first paragraph of the introduction feels disconnected from the core concept of the paper. It would benefit from a clearer link between the identified challenges and how NetAIF addresses these challenges specifically.
2. It would be helpful to add a summary of contribution at the end of the introduction.
3. The paper could benefit from a clear problem statement. Can the authors clarify what are the inputs to the proposed system? Additionally, can the authors confirm whether RL baselines in the paper are using equivalent input information to ensure a fair comparison?
4. In Table 1, adding noise and Hindsight Experience Replay (HER) appears to improve SAC's performance. Why were these enhancements not applied to PPO in Table 2? Since PPO seems to perform better in the experiments, why wasn't PPO tested in Table 1 for consistency?
5. What do the authors think about the comparison between the proposal and a model-based RL approach?

---

> ### Author Response · Authors · 2024-11-18
>
> We thank the reviewer for their thoughtful comments and for recognizing the novelty and practicality of NetAIF, particularly its ability to learn in real time without extensive pre-training. Below, we address the specific weaknesses and questions raised, and outline our plan to improve the manuscript.
>
> First, we acknowledge that the paper may be challenging for readers unfamiliar with Active Inference (AIF). To address this, we will expand the explanation of how NetAIF differs from standard neural networks, particularly its weight update mechanism, which leverages random attractor dynamics rather than traditional backpropagation. This expanded explanation will include how feedback control loops are used to minimize prediction error, stabilize the system, and dynamically adjust weights during operation. These clarifications will make the framework more accessible to a broader audience, including readers from a general machine learning background.
>
> Regarding the lack of a related work section, we appreciate the reviewer’s suggestion and will add a thorough discussion of prior research. This will include comparisons to neural network architectures, reinforcement learning (RL) methods, and other AIF-based frameworks. By situating NetAIF within the context of existing literature, we will highlight its unique contributions, particularly its ability to avoid pre-training, operate in real-time, and achieve significant computational efficiency.
>
> The reviewer raised an important question about whether NetAIF has access to privileged information compared to model-free RL algorithms. We confirm that NetAIF uses the same input data as the RL baselines for both training and execution. This ensures that all comparisons in the paper are fair. We will explicitly state this in the manuscript to avoid any ambiguity. Additionally, we will clarify why certain enhancements, such as adding noise and Hindsight Experience Replay (HER), were not applied to PPO in Table 1. The choice to limit enhancements in some cases was made to focus on baseline algorithm performance; however, we acknowledge the value of consistency and will discuss how these enhancements might impact results in future experiments.
>
> We also appreciate the reviewer’s comment about improving the introduction and providing a clearer link between the identified challenges and NetAIF’s contributions. We will revise the first paragraph of the introduction to better contextualize the challenges in robotic control and explain how NetAIF addresses these issues, particularly in dynamic and resource-constrained environments. Furthermore, we will include a summary of contributions at the end of the introduction to clearly outline the novelty and impact of the work.
>
> Finally, the reviewer raises a valuable point about comparing NetAIF with model-based RL approaches. While this paper focuses on comparisons with model-free RL, we recognize the importance of benchmarking against model-based methods, as they often excel in environments where dynamics are well-defined. We will add a discussion in the future work section about extending comparisons to include model-based RL methods, emphasizing NetAIF’s adaptability and scalability in diverse settings.
>
> In summary, we will revise the manuscript to address these points, improving accessibility, adding context, clarifying the experimental design, and strengthening the presentation of NetAIF’s contributions. Thank you for your valuable feedback, which has helped us refine and strengthen our work.

---

> > ### Comment · Reviewer_TcU3 · 2024-11-22
> >
> > Thank you for the rebuttal. I will maintain my evaluation for now. I believe the paper would benefit from a major revision to enhance its quality and impact.

---

### Official Review · Reviewer_D1VS · 2024-11-02

**Soundness:** 2
**Presentation:** 1
**Contribution:** 2
**Rating:** 3
**Confidence:** 4

**Summary:**

This paper proposes a novel method based on the active inference framework to solve a value-turning robotic manipulation task. They explain the active inference framework at a high-level, discuss their approach NetAIF, and compare it to a number of deep RL (DRL) approaches. They show an improvement in sample efficiency and performance compared to these competing methods on the considered task.

**Strengths:**

- Considering alternative algorithms to DRL which are more sample-efficient for specific applications is a timely and important problem.
- The evaluation does support the claims of sample efficiency and improved performance of the proposed algorithm.

**Weaknesses:**

- While the proposed algorithm, NetAIF, appears very interesting, it is very difficult to ascertain how it works from the exposition in the text. There is an Algorithm box, but it is missing many critical pieces of information, including how the new weight values are set and over what the prediction errors are computed. The paper would be significantly strengthed if rewritten to include more information about the learning rule and what sets the network architecture apart from a linear RNN.
- While I know the point of the paper was to evaluate the algorithm on a specific task, it makes it very hard to determine if this approach is scalable to other problems. I greatly appreciate the inclusion of a real robotic platform. However, for a machine learning conference, the paper would be significantly strengthened by evaluating over more benchmarks to determine the generalizability of the proposed method.

**Questions:**

- What exactly is the network architecture? How does the learning rule work?
- How well does the proposed method work on alternative benchmark tasks?

---

> ### Author Response · Authors · 2024-11-18
>
> We thank the reviewer for their thoughtful and constructive feedback and for recognizing the potential of our method as a sample-efficient alternative to DRL with significant performance improvements. Below, we address the concerns and questions raised and outline the steps we will take to improve the manuscript.
>
> First, we acknowledge the reviewer’s concern about the clarity of the network architecture and learning rule. To address this, we will provide a detailed explanation of the network structure, including its layer configuration, input/output specifications, and the role of feedback loops in stabilizing the system and enabling random attractor dynamics. Furthermore, we will clarify the learning rule by describing how prediction errors (calculated as the difference between the desired and current states) are used to guide weight updates. Unlike traditional backpropagation, NetAIF employs local feedback mechanisms and attractor dynamics to ensure stability and adaptability in real time. These additions will make the framework more accessible, particularly for readers unfamiliar with Active Inference or feedback-based learning systems.
>
> The reviewer also raised an important point about the generalizability of NetAIF beyond the valve-turning task. While this paper focuses on valve manipulation as a representative benchmark, we emphasize that the task is far from trivial. It requires precision, adaptability, and robustness under varying physical conditions such as friction and resistance. These variations make it an ideal testbed for evaluating NetAIF. Nonetheless, we agree that additional benchmarks would strengthen the generalizability of the method. While these were beyond the scope of the current study, we plan to extend NetAIF to more dynamic tasks, such as high-speed robotic manipulation, multi-agent systems, and quadcopter control, in future work. We will expand the manuscript’s discussion section to outline these directions and emphasize NetAIF’s potential for broader applications.
>
> We also appreciate the reviewer’s suggestion to compare NetAIF with model-based RL approaches. While this paper focuses on model-free RL baselines for fair comparisons of computational efficiency and adaptability, we recognize the value of benchmarking against model-based methods, which explicitly leverage system dynamics. In future studies, we plan to include such comparisons to further highlight NetAIF’s advantages in environments with unknown or changing dynamics. We will add a discussion in the future work section to address this point.
>
> Finally, we acknowledge the concern about the Algorithm box lacking sufficient detail on how weight values are set and how prediction errors are computed. We will revise the algorithm to include a step-by-step explanation of the initialization and weight adjustment process, an explicit definition of prediction error, and a description of how feedback loops and attractor dynamics ensure system stability. These revisions will make the methodology more transparent and reproducible.
>
> In summary, we will revise the manuscript to provide detailed explanations of the network architecture and learning rule, improve the algorithm description, and enhance the discussion of generalizability and comparisons to model-based RL. These updates will address the reviewer’s concerns and improve the clarity, rigor, and scope of the paper. We appreciate the valuable feedback, which has helped us identify key areas for improvement.

---

### Official Review · Reviewer_tE2D · 2024-11-04

**Soundness:** 2
**Presentation:** 2
**Contribution:** 2
**Rating:** 3
**Confidence:** 4

**Summary:**

The authors propose an algorithm to address some of the shortcomings of Deep Reinforcement Learning (DRL) algorithms such as : generalizability to changing/dynamic environment and compute efficiency. They tackle the task of automated valve-turning by formulating the problem from an Active Inference (AI) perspective which is built on Free Energy Principle (FEP). The results show significant improvement in task accuracy and compute efficiency over several DRL baselines.

**Strengths:**

The paper is well motivated and proposes an interesting approach to tackle an important issue in robotics.

**Weaknesses:**

The biggest weakness in this paper is the lack of experimental details. My questions in the following paragraph are all primarily clarifying several missing information that might be very useful for a reader.

**Questions:**

While the results comparing the task accuracy as well as compute efficiency to DRL methods seems very promising, In my opinion, several aspects of the paper needs to be improved.

Starting with experimental details : The paper seems to lack sufficient details about the different experimental set-up for a reader to reproduce the results. For example, what is the state space of the robot for this task? I don’t recollect it ever being mentioned in the entire paper, neither in the supplementary material.

What exactly is the prediction error for the valve turning task? For the target following task? Is it the difference between the current robot joint angles and the desired robot joint angles?

For the DRL baselines, what is the state and action space? What reward function is used?
Section 2.2 mentions a “random attractor”, how does this fit into the Algorithm 1?

The video supplement is helpful to gain context about the task and the robot motion is impressive. However, since the main goal of the approach was to show that it can adapt to changes in the environment, why not show valve turning for the different shapes or with different frictional properties? Are there any failure cases? If so, why? It might also be helpful to see how the DRL policies look like in real world, are they so bad that deploying them was not feasible?

In figure 3, For the valve turning task, what are the external, internal and blanket states? Providing this information could give the reader some context and improve readability of the paper

---

> ### Author Response · Authors · 2024-11-18
>
> We thank the reviewer for their thoughtful comments and for acknowledging the importance of the problem tackled in this paper, as well as the promise shown by NetAIF in improving task accuracy and computational efficiency compared to DRL methods. Below, we address the key concerns and questions raised to improve the clarity and rigor of the manuscript.
>
> First, we recognize the reviewer’s concern about the lack of experimental details. To address this, we will revise the manuscript to include a detailed description of the experimental setup, including the state space of the robot and the action space. For the valve-turning task, the state space includes the joint positions, velocities, and end-effector pose (position and orientation), while the action space corresponds to continuous joint velocity commands. Additionally, we will clarify the reward function used for the DRL baselines, which was designed to minimize the deviation from the target valve rotation angle while penalizing unnecessary motion. Providing these details will ensure that readers have all the information needed to reproduce the results.
>
> The reviewer also asked for a precise definition of prediction error in the context of the valve-turning task. We will add an explicit explanation to the manuscript, defining prediction error as the difference between the current robot joint angles and the desired joint angles required to achieve the target valve rotation. This metric is central to evaluating NetAIF’s performance and will be clarified to avoid ambiguity.
>
> Regarding the concept of the random attractor mentioned in Section 2.2, we will expand the discussion to explain how this mechanism fits into Algorithm 1. Specifically, the random attractor enables the system to explore state spaces efficiently while maintaining stability through feedback loops. This mechanism is integral to NetAIF’s ability to adapt dynamically to changes in the environment, and we will ensure that this is clearly explained.
>
> We also appreciate the reviewer’s suggestion to provide more context on the external, internal, and blanket states mentioned in Figure 3. We will include additional details to explain how these states interact in the framework of Active Inference and how they are operationalized in the valve-turning task. This will enhance the reader’s understanding of the theoretical foundations and their practical implementation.
>
> Finally, the reviewer raised a valid point about the video supplement and the lack of demonstrations of failure cases or extreme scenarios, such as varying frictional properties or valve shapes. To address this, we will expand the supplementary video to include additional test cases, such as high-friction or slippery surfaces, and examples of failure modes. This will highlight NetAIF’s adaptability and robustness in diverse and challenging conditions, as well as provide insights into the limitations of DRL policies under the same scenarios.
>
> In summary, we will revise the manuscript to address these concerns by providing detailed descriptions of the experimental setup, prediction error, and random attractor dynamics, as well as expanding the supplementary material to include diverse test cases. These revisions will enhance the clarity, rigor, and completeness of the paper. We sincerely thank the reviewer for their constructive feedback, which has been invaluable in identifying key areas for improvement.

---

> > ### Comment · Reviewer_tE2D · 2024-11-27
> >
> > Thanks for rebuttal. Im going to retain my rating.
> > 1) I don't see any changes to the supplementary video that the authors said they'll add. On closer inspection of the video, around ~1.40 - 1.55 min mark in the video, does the robot turn the valve just 45 degrees? if you look close enough it actually seems like the robot gripper sometimes just rotates without actually turning the valve. This brings into question - how does the prediction error capture the act of turning the valve? Clearly writing the mathematical equation of the error calculation could be very helpful.
> >
> > 2) Experimental details of the baseline RL algorithms are still missing. What was the reward function used?
> >
> > 3) The state space is still not clear, for example, how does the robot know that the valve has turned and the task is complete?
> >
> > There is plenty of room for improvement.

---

### Note · Program_Chairs · 2025-01-15
**Submission Desk Rejected by Program Chairs**

The paper is a dual submission with 12631.